# Group B *Streptococcus*-Induced Macropinocytosis Contributes to Bacterial Invasion of Brain Endothelial Cells

**DOI:** 10.3390/pathogens11040474

**Published:** 2022-04-15

**Authors:** Eric R. Espinal, Teralan Matthews, Brianna M. Holder, Olivia B. Bee, Gabrielle M. Humber, Caroline E. Brook, Mustafa Divyapicigil, Jerod Sharp, Brandon J. Kim

**Affiliations:** 1Department of Biological Sciences, University of Alabama, 300 Hackberry Lane, Tuscaloosa, AL 35405, USA; erespinal@crimson.ua.edu (E.R.E.); tamatthews@crimson.ua.edu (T.M.); bmholder@crimson.ua.edu (B.M.H.); gmhumber@crimson.ua.edu (G.M.H.); cebrook@crimson.ua.edu (C.E.B.); mdivyapicigil@crimson.ua.edu (M.D.); sjsharp2@crimson.ua.edu (J.S.); 2Department of Chemical and Biological Engineering, University of Wisconsin, Madison, WI 53706, USA; obee@ulm.vcom.edu; 3Department of Microbiology, Heersink School of Medicine, University of Alabama at Birmingham, Birmingham, AL 35294, USA; 4Center for Convergent Biosciences and Medicine, University of Alabama, Tuscaloosa, AL 35487, USA; 5Alabama Life Research Institute, University of Alabama, Tuscaloosa, AL 35487, USA

**Keywords:** blood–brain barrier, bacterial meningitis, group B *Streptococcus*, macropinocytosis, iPSC-BEC, host–pathogen interaction

## Abstract

Bacterial meningitis is defined as serious inflammation of the central nervous system (CNS) in which bacteria infect the blood–brain barrier (BBB), a network of highly specialized brain endothelial cells (BECs). Dysfunction of the BBB is a hallmark of bacterial meningitis. Group B *Streptococcus* (GBS) is one of the leading organisms that cause bacterial meningitis, especially in neonates. Macropinocytosis is an actin-dependent form of endocytosis that is also tightly regulated at the BBB. Previous studies have shown that inhibition of actin-dependent processes decreases bacterial invasion, suggesting that pathogens can utilize macropinocytotic pathways for invasion. The purpose of this project is to study the factors that lead to dysfunction of the BBB. We demonstrate that infection with GBS increases rates of endocytosis in BECs. We identified a potential pathway, PLC-PKC-Nox2, in BECs that contributes to macropinocytosis regulation. Here we demonstrate that downstream inhibition of PLC, PKC, or Nox2 significantly blocks GBS invasion of BECs. Additionally, we show that pharmacological activation of PKC can turn on macropinocytosis and increase bacterial invasion of nonpathogenic yet genetically similar *Lactococcus lactis*. Our results suggest that GBS activates BEC signaling pathways that increase rates of macropinocytosis and subsequently the invasion of GBS.

## 1. Introduction

Bacterial meningitis is a life-threatening disease characterized by inflammation of the meninges that surround the central nervous system (CNS). Meningitis has a disproportionate effect on children throughout the world, especially under 5 years old [1]. Due to vaccination efforts and the development of antibiotic therapy, the rates of mortality and morbidity have drastically decreased. However, patients recovering from meningitis are prone to neurological sequelae such as cognitive impairment [2]. Developing countries continue to have slightly higher rates of morbidity; therefore, there is an effort to understand the mechanisms behind bacterial invasion of the brain. While not necessary for disease, bacteremia, the presence of bacteria in the circulating bloodstream, increases the likelihood of meningitis [3]. The bacteria must then travel to, interact with, and penetrate the blood–brain barrier (BBB) that surrounds the CNS to cause damage.

Group B *Streptococcus *(GBS), also known as *Streptococcus agalactiae,* is a Gram-positive organism and is the leading cause of neonatal meningitis [4]. GBS can reside asymptomatically in the vaginal tract of about 30% of women [5]. Therefore, most women are screened during pregnancy and given antibiotics as prophylaxis if they are carriers. Understanding the mechanisms behind infection is paramount to developing novel therapeutics. Therefore, efforts have been made in identifying several virulence factors of GBS that enhance pathogenicity such as expression of lipoteichoic acids and pilus components for cell adhesion and invasion [6,7].

The BBB is composed of highly specialized brain microvascular endothelial cells (BECs) that serve to protect the CNS [8,9]. These BECs differ from peripheral endothelial cells in that they express a variety of tight junction proteins and efflux transporters. In addition, BECs have lower rates of endocytosis to maintain homeostasis in the CNS free of pathogens and other toxins [10]. A hallmark of bacterial meningitis is disruption of these BBB properties that allow bacteria to translocate through the BBB. Previous work examining GBS has observed bacteria inside membrane-bound vesicles, suggesting that GBS may utilize endocytosis as a route of entry [11,12]. In addition, transmission electron microscopy (TEM) reveals what looks like engulfment of the bacteria with the plasma membrane of the BEC, suggesting macropinocytosis as a potential mechanism for invasion [11].

Macropinocytosis is the nonselective uptake of extracellular molecules [13]. Macropinocytosis is an actin-dependent process characterized by plasma-membrane ruffling, formation of a macropinosome, and finally internalization of extracellular material [13]. Previous studies have shown that inhibition of actin-dependent processes such as cytoskeletal rearrangement decreases macropinocytosis and bacterial invasion of BECs [11,14]. While macropinocytosis is tightly regulated at the BBB, pathogens have demonstrated the ability to utilize this mechanism for entry [14,15]. However, whether GBS utilizes macropinocytosis as a mechanism for bacterial invasion of BECs remains unknown.

Here we find that the PLC-PKC-Nox2 pathway in BECs contributes to macropinocytosis regulation. Previous studies have shown that PLC and PKC activation are important for macropinosome formation [15,16,17]. In addition, PKC can activate Nox2 to promote membrane ruffling and fluid-phase uptake [17,18]. Therefore, we sought to investigate the role of this pathway in GBS invasion of BECs. We use an induced pluripotent stem-cell (iPSC) model to differentiate stem cells into brain microvascular endothelial-like cells (iPSC-BECs) [19,20]. We and others have utilized the iPSC-BECs as a model to study viral and bacterial BBB interaction, including with GBS [4,21,22,23,24,25]. We demonstrate that GBS invasion of BECs increases global rates of endocytosis. In addition, downstream pharmacological inhibition of PLC, PKC, or Nox2 significantly blocks GBS invasion of BECs. Conversely, activation of PKC can turn on macropinocytosis and increase invasion of a nonpathogenic yet genetically similar organism, *Lactococcus lactis*. This study highlights a regulatory pathway of macropinocytosis for bacterial invasion of BECs.

## 2. Results

### 2.1. GBS Infection Increases Rates of Endocytosis

Previous work has demonstrated that GBS has been observed in membrane-bound vesicles inside BECs [11]. This observation suggests that GBS utilizes endocytic mechanisms to invade BECs. Therefore, we investigated how GBS infection affects endocytosis rates of BECs, which are normally tightly regulated. We utilized our iPSC-derived BEC-like cells to measure relative uptake of dextran with and without infection (Appendix A). BECs were treated with a fluorescent 10k-Da dextran and either mock-infected or infected with GBS for 5 h, a time point where we have previously observed BBB disruption [9,26]. Cells were washed then lysed with RIPA buffer and measured for relative fluorescent emission on a plate reader. We observed a significant increase in intracellular fluorescent dextran with GBS infection compared to uninfected cells (Figure 1A). This suggests that during infection GBS may turn on general endocytic pathways. Using fluorescence microscopy, we observed fluorescent dextran associated with the BECs after GBS infection (Figure 1B). Finally, dextran uptake was measured using flow cytometry and we observed an increased uptake of dextran compared to the controls (Figure 1C). Taken together, these results demonstrate that GBS may activate global endocytic mechanisms of BECs.

### 2.2. Inhibition of PLC Decreases GBS Invasion

Previous studies have shown that pathogens can utilize macropinocytosis as a form of entry in BECs [14]. Additionally, it has been demonstrated that blocking actin polymerization reduces GBS invasion of BECs, further motivating investigation into macropinocytotic pathways [11]. However, it is unclear which mechanisms regulate this actin-dependent process. Recruitment of phospholipase C (PLC) to its substrate phosphatidylinositol (4,5)-biphosphate (PIP2) at the plasma membrane has been implicated for its importance in actin cytoskeleton activation [27,28]. Previously, inhibition of PLC led to a decrease in the ability to form macropinosomes [16]. Therefore, we hypothesized that inhibition of PLC would inhibit macropinocytosis and lead to a decrease in bacterial invasion. We utilized a common inhibitor of PLC, ET 18-OCH_3_, to measure relative invasion of GBS [28]. We employed cell-association assays to determine adherence, and antibiotic-protection assays to determine invasion, where BECs were pretreated with various concentrations of ET 18-OCH_3_ (10, 100, and 1000 nM; 2 h) before infection with GBS. After incubation with ET 18-OCH_3_ we observed a decrease in the amount of bacteria associated to the BECs (Figure 2A). In addition, we saw a similar significant decrease in bacterial invasion in a dose-dependent manner compared to the vehicle (Figure 2B). However, in the immortalized hCMEC/D3 cells, we did not observe any decreases to adherence while we still saw a decrease in bacterial invasion (Appendix A). Importantly, we did not observe any cytotoxic effects to BECs or impairment to bacterial growth at concentrations used (Appendix A). We demonstrate that inhibition of PLC leads to a decrease in GBS invasion.

### 2.3. Inhibition of PKC Decreases Invasion of GBS

PLC hydrolysis of PIP2 at the plasma membrane releases the lipid secondary messenger diacylglycerol (DAG), which is an activator of Protein Kinase C (PKC). Previous studies have shown that PKC activation leads to plasma-membrane ruffling, a hallmark of macropinocytosis; therefore, we sought to determine the role of PKC in GBS invasion [15,17]. We utilized two common PKC inhibitors, Calphostin C and Rottlerin, and employed cell association and invasion assays to enumerate bacterial loads. BECs were first treated with Calphostin C (10 nM and 100 nM; 2 h) followed by infection with GBS. We observed no change in GBS adherence to BECs; however, GBS invasion significantly decreased in a dose-dependent manner using Calphostin C (Figure 3A,B). Similarly, we employed the same assay using the hCMEC/D3 cell line. Once again, we saw a significant decrease in bacterial invasion of BECs while there was no change in adherence (Appendix A). Next, we used Rottlerin as another inhibitor of PKC. We treated the BECs with Rottlerin (10, 100, and 1000 nM; 2 h) followed by GBS infection and observed a similar decrease in GBS invasion across all doses with no change in adherence. (Figure 3C,D). We validated these results using the hCMEC/D3 cells and observed no changes in adherence and a significant decrease in invasion (Appendix A). In addition, Calphostin C and Rottlerin had no effect on bacterial growth or caused cell death at the used concentrations (Appendix A). While pharmacological inhibitors may have off-target effects, overlapping results of two different inhibitors suggest effective inhibition of PKC. Taken together, these results suggest that PKC activation contributes to GBS invasion of BECs.

### 2.4. Nox2 Contributes to GBS Invasion of BECs

We have shown that PKC is an important regulator of macropinocytosis, leading to invasion of GBS. We then wanted to identify a potential downstream molecule that PKC could activate to contribute to a macropinocytotic pathway. Nox2 is a NADPH oxidase found in intracellular membranes that generates reactive oxygen species (ROS) used in redox signaling [18]. Nox2 is composed of several transmembrane and cytosolic subunits that come together to activate the complex. Importantly, PKC-mediated activation of the p47^phox^ cytosolic subunit is required to translocate to the membrane and activate Nox2 [29,30]. In previous studies, Nox2 inhibition leads to a decrease in fluid-phase uptake in macrophages. Conversely, activators of Nox2 exhibit characteristic macropinocytotic plasma-membrane ruffling [17,18]. Therefore, we sought to identify the role of Nox2 in macropinocytosis entry of GBS into BECs. We employed an adherence and invasion assay using diphenyliodinium (DPI) and Imipramine, common inhibitors of Nox2 [31]. BECs were treated with DPI (10, 100, and 1000 nM; 2 h) before infection, and antibiotic protection assays were conducted. While we saw no changes in adherence, inhibition of Nox2 lead to a significant decrease in bacterial invasion at the strongest concentration (Figure 4A,B). Interestingly, when we treated the hCMEC/D3 line with DPI, we saw a slight increase in adhesion but a still a significant decrease in invasion (Appendix A). In addition, BECs were similarly treated with Imipramine (1 and 10 μM; 2 h), and cell-adhesion and invasion assays were performed. Similar to DPI, we saw no changes in bacterial adherence; however, we observed a decrease in GBS invasion of BECs (Figure 4C,D). We then validated our results by treating hCMEC/D3 cells with Imipramine and observed a reduced ability to invade the BECs across both doses (Appendix A). Neither DPI nor Imipramine affected bacterial growth or caused cell cytotoxicity at concentrations used (Appendix A). Altogether, we identified a potential pathway that regulates macropinocytosis and contributes to bacterial invasion.

### 2.5. PMA Stimulates Macropinocytosis and Increases Bacterial Invasion

Phorbol 12-myristate 13-acetate (PMA) is a phorbol ester molecule known to stimulate macropinocytosis and initiate membrane ruffling [31]. PMA acts as an analogue to DAG to activate PKC. Therefore, we investigated its effect on macropinocytosis and bacterial invasion of BECs. We employed a dextran-uptake assay where we treated the cells with PMA for 5 h and added dextran to the media. After the incubation period, we used flow cytometry to quantify the amount of intracellular dextran. When treated with PMA, there was an increase in the amount of dextran within the BECs, corroborating previous results that PMA can induce macropinocytosis (Appendix A). However, it is unclear if this PKC activation can increase bacterial invasion of BECs. We treated BECs with PMA (0.1 and 1 μM; 2 h) followed by an adherence and invasion assay with GBS. However, there was no significant increase in bacterial invasion (Appendix A). Possibly, GBS is already effective in activating this macropinocytosis pathway contributing to invasion, so there may be little to no additional stimulation of PKC when using PMA. Therefore, we utilized the nonpathogenic, noninvasive Gram-positive bacterium *Lactococcus lactis*. *L. lactis* does not readily invade BECs, therefore we hypothesized that macropinocytosis stimulation with PMA may allow entry of the bacteria. Cells were treated with PMA and infected with *L. lactis* at an MOI of 50. We did not observe any changes in bacterial adherence to the BECs (Figure 5A). However, *L. lactis* had a significant increase in invasion of the BECs following PMA treatment (Figure 5B). Additionally, PMA did not induce any cytotoxic effects on the BECs or inhibit growth of either bacteria (Appendix A). We show that PKC can be stimulated with the phorbol ester PMA to induce macropinocytosis. In addition, turning on macropinocytosis can allow for entry of bacteria that are normally noninvasive into BECs.

## 3. Discussion

Previously, GBS has been found to invade BECs inside membrane-bound vesicles [11]. In addition, GBS has been shown to colocalize to autophagosomes and induce an autophagic response, whereas noninvasive mutants cannot induce this response, further suggesting the role of endocytosis as a mechanism of invasion. Looking at the intracellular fate of GBS after invasion, mutants with knockouts of autophagy-related proteins have been shown to increase the intracellular survival rate compared to WT GBS [32]. Efforts have been made to elucidate the mechanisms behind these endocytic pathways. In fact, we show that GBS infection increases global rates of endocytosis through the increased uptake of fluorescent dextran (Figure 1A–C). One option of bacterial invasion is through macropinocytosis, where the plasma membrane ruffles and engulfs GBS, which fits previous observation of membrane protrusions at sites of GBS attachment and dependency of actin polymerization in BECs [11]. However, regulators of this process remain unknown. Here we demonstrate one pathway, PLC-PKC-Nox2, that regulates activation of macropinocytosis and contributes to bacterial invasion.

During infection with GBS, PLC may be activated to turn on a signaling pathway that regulates macropinocytosis, including the downstream effectors PKC and Nox2. Pharmacological inhibition of PLC with ET 18-OCH_3_ showed a decrease in GBS invasion in a dose-dependent manner (Figure 2B). PKC, a cytosolic kinase, has been implicated in membrane ruffling [17]. Therefore, we demonstrated that inhibition of PKC with two common inhibitors, Calphostin C and Rottlerin, leads to a decrease in bacterial invasion (Figure 3B,D). Inactivation of PKC presumably turns off its ability for phosphorylation. In addition, we investigated the role of Nox2 as a downstream effector of macropinocytosis. Using Nox2 inhibitors, DPI and Imipramine, we observed a decrease in bacterial invasion (Figure 4B,D). Nox2 serves as a NADPH oxidase that produces ROS for redox signaling. Previous studies have shown that overproduction of ROS in other cell types activate the actin-binding protein cofilin to stimulate actin reorganization [18]. Inactivation of Nox2 decreases the production of ROS, leading to a decrease in this redox signaling that regulates actin. Once inside the cell, it has been shown that the intracellular survival of GBS begins to decline after 6 h [9]. However, we are unsure of the fate of GBS inside the cell after pharmacological inhibition of this pathway at different points. Future work of this project involves examining the cellular trafficking and intracellular survival of GBS after infection and inhibition of PLC, PKC, or Nox2. In some cases, we observed that GBS adherence remained the same while invasion decreased across pharmacological inhibition. This may be due to the fact that GBS possesses an array of virulence factors that specifically mediate either adherence or invasion or both. Therefore, it is possible that blocking a pathway utilized for invasion will have no effect on bacterial adherence. In addition to our iPSC-BEC model, results were corroborated in a well-established immortalized BEC model, further validating utilization of the iPSC-BEC model in studying host–pathogen interactions of the BBB (Appendix A) [10]. Taken together, we propose that the PLC-PKC-Nox2 pathway is a regulator of macropinocytosis and GBS invasion of BECs.

While we observe this pathway as a modulator of macropinocytosis for GBS invasion of BEC, it is also possible that GBS utilizes a variety of other endocytic mechanisms for virulence. This possibility is evidenced in the fact that there are still observed invasions even after pharmacological inhibition and not completely abolished. In fact, other endocytic mechanisms have been previously investigated for GBS invasion of BEC in other models. For example, it has been shown that host cytosolic phospholipase A_2_α (cPLA_2_α) increases cysteinyl leukotrienes (CysLTs) after GBS infection, which in turn activate PKCα to aid in invasion [33]. While it has also been discovered that sphingosine 1-phosphate (S1P) binds its receptor to activate an epidermal growth-factor receptor (EGFR) upstream of cPLA_2_α and CysLTs [34]. In addition, GBS has been shown to bind host integrins, specifically α5β1 and αvβ3, through the adhesin Srr2 [35,36,37]. Finally, endocytosis is known to come in a variety of forms other than macropinocytosis, such as clathrin-mediated and caveolin-mediated endocytosis [38]. When dendritic cells are treated with inhibitors of clathrin-mediated internalization, GBS invasion is significantly reduced. In addition, inhibition of the large GTPase dynamin also decreases GBS internalization [39]. On the other hand, depletion of lipid rafts on the cell surface reduces bacterial invasion in a variety of cell types [14,39,40]. Of note, when macrophages were treated with clathrin-mediated and caveolin-mediated inhibitors, internalization of FITC-dextran after PMA stimulation was not affected [18]. In addition, macrophages treated with the Nox2 inhibitor Imipramine did not affect clathrin-mediated or caveolin-mediated endocytosis of transferrin or albumin, respectively [31]. Furthermore, activation of PKC through PMA has been shown to induce membrane ruffling and macropinosomes formation, characteristics of macropinocytosis; conversely, inhibition of PLC, PKC, or Nox2 reduces the number of macropinosomes and membrane ruffles [16,17,18,31]. Altogether, while we cannot exclude other forms of endocytosis as a mechanism for invasion, these results suggest that PLC-PKC-Nox2 is a regulator of macropinocytosis that GBS utilizes for invasion.

It is still unknown what bacterial factor may trigger the activation of PLC. Interestingly, a nonencapsulated mutant of the same type III strain exhibited a hyperinvasive phenotype compared to the capsulated WT COH1 [11]. It is hypothesized that this may be due to the capsule’s ability to occlude some other virulence factors from direct cell interaction. It has been shown that cholesterol and lipid rafts in the membrane are important for invasion of pathogens via macropinocytosis [14]. Therefore, potentially, GBS interacts with lipid rafts at the membrane surface to stimulate downstream effectors such as PLC. In addition, it is unknown if activation of cofilin occurs in BECs similar to macrophages [18]. While the iPSC-BEC model is well-suited for studying BBB interactions in vitro, a limitation of the study is understanding whether this pathway holds true in vivo. The BBB benefits from interactions with astrocytes and pericytes that are difficult to recapitulate in vitro [41]. Astrocytes and pericytes surround the BBB to form a neurovascular unit and aid in upregulating its specific properties such as tight junctions and polarized expression of transporters [41]. Previously, GBS has been shown to invade astrocytes and induce the secretion of proinflammatory cytokines, which leads to further disease progression [12]. Therefore, in vivo models offer insight into how these pathways contribute to the dysfunction of the neurovascular unit as a whole.

## 4. Materials and Methods

### 4.1. Bacterial Strains and Cell Lines

Group B *Streptococcus* (GBS) wild-type (WT) strain COH1 (Serotype III, multilocus sequence type 17) was used in infection experiments [42]. GBS was grown overnight before infection experiments in Todd-Hewitt Broth (THB) at 37 °C. *Lactococcus lactis* WT was also used and was grown in THB overnight at 30 °C. Induced pluripotent stem cells (iPSCs) were cultured onto Matrigel (Corning, NY, USA)-coated 6-well plates (Corning) in StemFlex (Gibco) medium, changed daily. Cells were passaged twice a week as needed. Immortalized hCMEC/D3s were cultured onto 1% rat-tail collagen (VWR) in EndoGro MV media (Millipore). Cells were passed when necessary and seeded onto 24-well plates (Corning).

### 4.2. Brain Endothelial-Cell Differentiation

iPSC-derived brain endothelial cells (BECs) were properly differentiated each week according to the protocol outlined in Stebbins et al., 2016 [19,20]. Briefly, iPSCs in a single cell suspension were seeded onto Matrigel (1 mg in 12 mL DMEM/F12 (Gibco, Waltham, MA, USA))-coated flasks (Corning) at 10,000 cells/cm^2^ for expansion over 3 days in StemFlex, with medium changed daily. After 3 days, differentiation into BECs was initiated with the use of unconditioned medium (UM), changed daily for 6 days. BECs were then expanded with EC medium, supplemented with 1% B27 (Gibco), 20 ng/mL basic fibroblast growth factor (bFGF) (PeproTech, Hamburg, Germany ), and 10 μM retinoic acid (RA) (Sigma, Burlington, MA, USA) for 2 days. Differentiated BECs were purified the next day and seeded onto collagen IV (Sigma), fibronectin (Sigma), and water-coated plates (4:1:45) and transwell inserts (4:1:5) (Corning) at a seeding density of 500,000 cells/well for a 24-well plate. Media was changed the next day to EC medium without bFGF and RA. BECs were validated by measuring transendothelial electrical resistance (TEER) on D9 and D10 and by immunostaining for BEC expression markers on D10.

### 4.3. Bacterial Preparation and Infection Assays

From the overnight stock, bacteria were subcultured in THB at 37 °C and grown until the optical density reading at 600 nm (OD600 nm) reached 0.4–0.6. Bacteria were then spun down and resuspended in phosphate buffer solution (PBS) until OD600 nm reached 0.4. Finally, bacteria were diluted 1:10 in EC medium. The purified BECs cultured onto collagen IV/fibronectin-coated plates were infected at a multiplicity of infection (MOI) of 10, unless noted otherwise. The adherence and invasion assays were performed according to protocols previously described [7,9,11,43]. Briefly, the adherence plates were incubated for 30 min at 37 °C + 5% CO_2_. The wells were then washed 5 times with PBS and lysed in 0.025% Triton X-100. Serial dilutions were performed followed by plating onto THB plates to quantify adherent bacteria. The invasion plates were incubated for 2 h at 37 °C + 5% CO_2_ and washed 3 times with PBS. Cells were then incubated with gentamicin (100 μg/mL) for 2 h and washed 3 times with PBS. Finally, they were lysed with 0.025% Triton X-100 and plated onto THB plates using a spread-plate method. For dextran-uptake assays, cells were treated with dextran (1 mg/mL) diluted in EC medium without bFGF and RA and infected simultaneously with GBS for 5 h at 37 °C + 5% CO_2_. Cells were then lysed in RIPA buffer, and relative fluorescence was measured with a plate reader.

### 4.4. Flow Cytometry

To quantify the amount of intracellular dextran in BECs, cells were treated with dextran and infected either with or without GBS for 5 h. Cells were then washed 1× with PBS and 200 μL of trypsin for 10 min at 37 °C + 5% CO_2_. They were transferred to a 15 mL conical tube, which was then spun down and resuspended in PBS 2×. After spinning for the last time, cells were resuspended in 500 μL of 1% FBS in PBS. Cells were then filtered and run through an Attune NxT flow cytometer.

### 4.5. Inhibitors and Antibodies

Calphostin C was obtained from Cayman Chemicals (Ann Arbor, MI, USA). Phorbol 12-myristate 13-acetate (PMA) was purchased from InvivoGen (San Diego, CA, USA). Diphenyleneiodonium chloride (DPI) and Imipramine were purchased from Sigma-Aldrich (St. Louis, MO, USA). Rottlerin and ET 18-OCH3 were purchased from Tocris (Minneapolis, MN, USA). Dextran was obtained from Thermo (Waltham, MA, USA) and VWR (Radnor, PA, USA). Antibodies were purchased as previously described [9,20].

### 4.6. Live/Dead Assays

Cells were incubated with inhibitors to test for cytotoxic effects. Inhibitors were diluted at various concentrations in either EC medium without bFGF and RA for iPSC-BECs or EndoGro MV medium for immortalized hCMEC/D3s for 5 h. After incubation, cells were trypsinized and mixed 1:1 with Trypan Blue. Live and dead cells were counted on a hemacytometer, and figures were represented as percentage of cells alive.

### 4.7. Growth Curves

Bacteria were incubated with respective inhibitors to test for bactericidal or bacteriostatic effects. Inhibitors were diluted in Todd-Hewitt Broth (THB) at varying concentrations and infected with COH1 WT. Infected broth was then transferred into a 96-well plate (VWR). Absorbance was read at OD600 nm in a SpectraMax ID3 (Molecular Devices, San Jose, CA, USA) plate reader at 37 °C every 30 min intervals overnight.

### 4.8. Statistics

GraphPad Prism version 9.0.0 was used for all statistical analysis. For pairwise comparisons, a 2-tailed Student’s *t*-test was used where appropriate. For multiple comparisons, a one-way analysis of variance (ANOVA) was used. Outliers were identified using GraphPad’s analysis and excluded from the data set. Statistical significance was accepted at a *p* value of less than 0.05.

## Figures and Tables

**Figure 1 pathogens-11-00474-f001:**
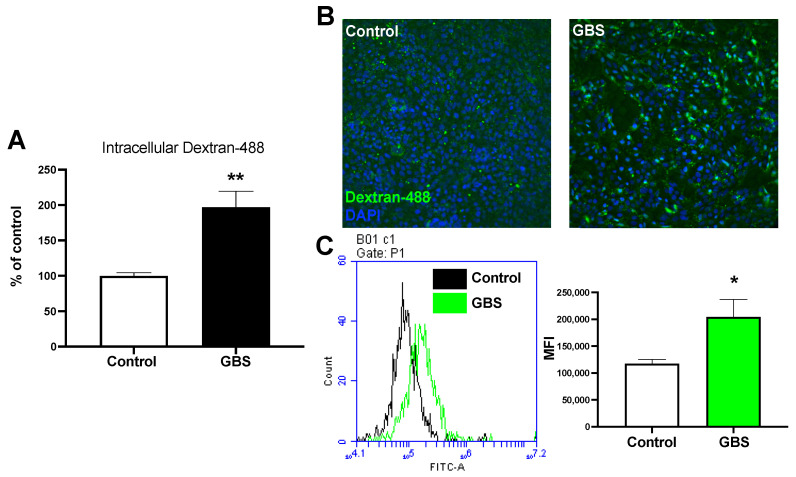
GBS infection of BECs increase global rates of endocytosis. (**A**) BECs take up fluorescent dextran during GBS infection for 5 h at an MOI of 10. (**B**) Fluorescence microscopy was used to visualize the increase in fluorescent 10 kD-dextran in cells infected with GBS. Cell nuclei were stained with DAPI (blue) with surrounding dextran (green) (10×). (**C**) Relative fluorescence intensity from uptake of fluorescence 10 kD-dextran measured through flow cytometry between GBS-infected and mock-infected cells. All experiments were conducted in triplicate. Error bars represent SD. Student’s *t* test was used to determine significance. * *p* < 0.05; ** *p* < 0.01.

**Figure 2 pathogens-11-00474-f002:**
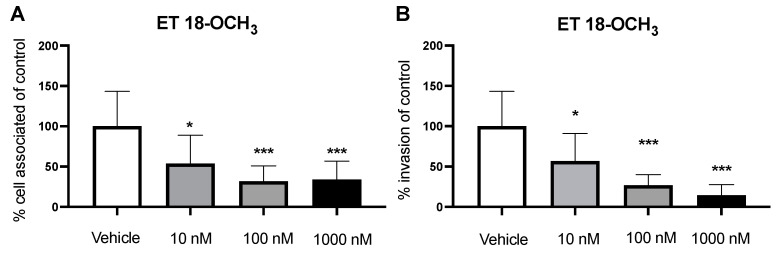
Inhibition of PLC leads a decrease in GBS adherence and invasion. Cells preincubated for 2 h with various concentrations of ET 18-OCH_3_ followed by GBS infection. (**A**,**B**) Adherence and invasion of GBS on BECs after treatment with ET 18-OCH_3_. Data are presented as mean values from three independent iPSC-derived BEC differentiations conducted in triplicate (*n* = 9). Error bars represent SD. ANOVA was used to determine the significance across the different conditions. * *p* < 0.05; *** *p* < 0.001, versus vehicle.

**Figure 3 pathogens-11-00474-f003:**
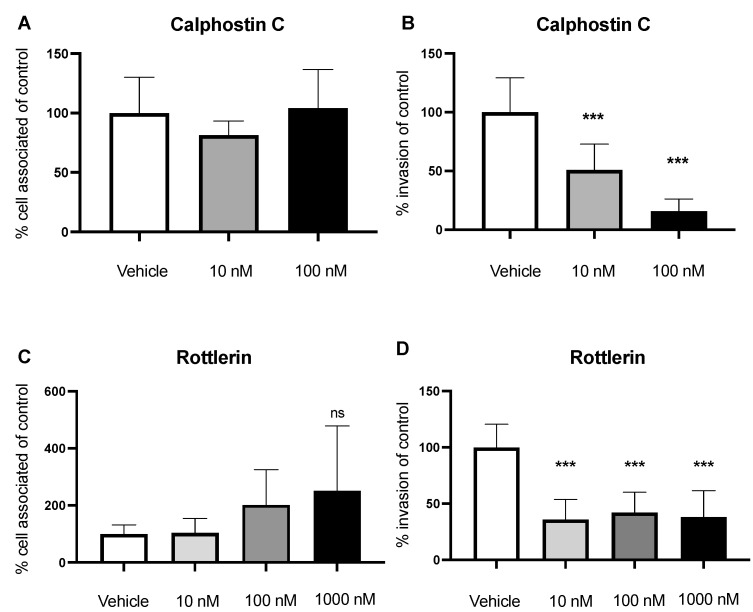
PKC inhibition shows a decrease in GBS invasion. (**A**) GBS adherence and (**B**) invasion were measured after BECs were treated with Calphostin C. BECs treated with Rottlerin to investigate (**C**) adherence and (**D**) invasion, and we observe a decrease in invasion. Data are presented as mean values from three independent iPSC-derived BEC differentiations conducted in triplicate (*n* = 9). One outlier was excluded at 10 nM for the invasion assay of Rottlerin (*n* = 8). Error bars represent SD. ANOVA was used to determine the significance across the different concentrations. *** *p* < 0.001, versus vehicle; ns: not significant.

**Figure 4 pathogens-11-00474-f004:**
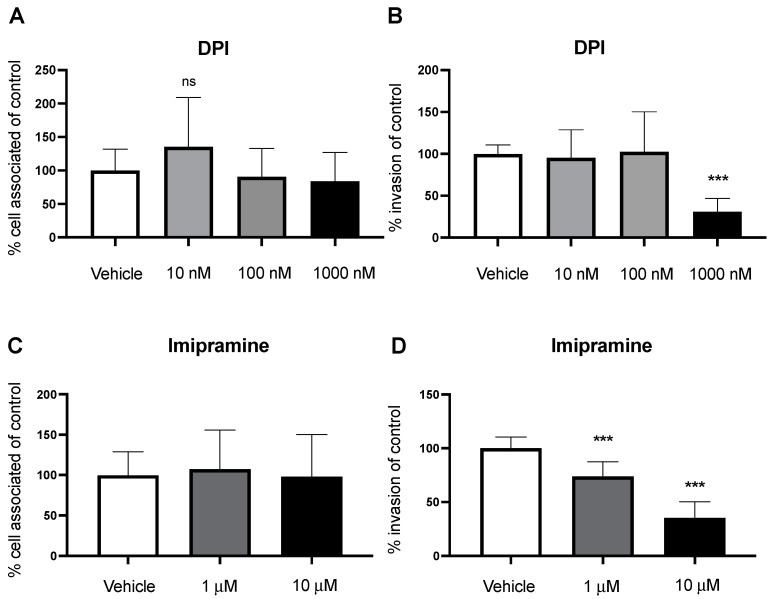
Nox2 inhibition leads to a decrease in bacterial invasion. Pharmacological inhibition of Nox2 was performed with DPI at various concentrations to observe (**A**) adherence and (**B**) invasion. (**C**,**D**) BECs were pretreated with Imipramine at 1 and 10 μM to measure adherence and invasion. It was observed that Nox2 inhibition decreases GBS invasion of BECs. Data are presented as mean values from three independent iPSC-derived BEC differentiations conducted in triplicate (*n* = 9). Adhesion data for Imipramine (**C**) are presented as mean values from three independent iPSC-derived BEC differentiations conducted in duplicate (*n* = 6). Error bars represent SD. ANOVA was used to determine the significance across the different concentrations. *** *p* < 0.001, versus vehicle; ns: not significant.

**Figure 5 pathogens-11-00474-f005:**
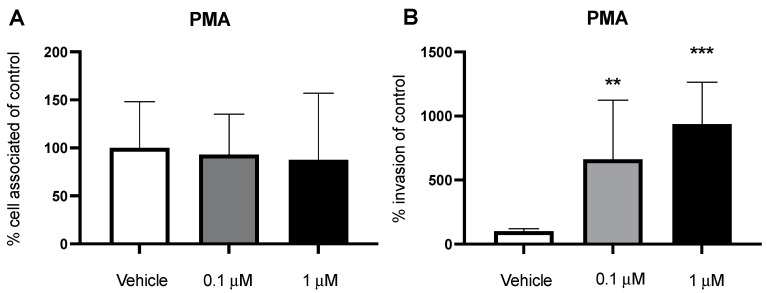
PMA stimulates an increase of a non-invasive *L. lactis.* BECs were preincubated at 0.1 μM and 1 μM for 2 h of PMA. (**A**,**B**) BECs were infected with *L. lactis* to assess adherence and invasion. PMA increases the invasion of *L. lactis*. Data are presented as mean values from three independent iPSC-derived BEC differentiations conducted in triplicate (*n* = 9). One outlier was excluded at 1 μM for the invasion assay (*n* = 8). Error bars represent SD. ANOVA was used to determine the significance across the different concentrations. ** *p* < 0.01; *** *p* < 0.001, versus vehicle.

## Data Availability

Data are contained within the article or Appendix A.

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
