# Peer review of "Group B Streptococcus-Induced Macropinocytosis Contributes to Bacterial Invasion of Brain Endothelial Cells"

_pathogens, 2022, doi:10.3390/pathogens11040474_

Round 1

Reviewer 1 Report

This is an interesting manuscript investigating the role of macropinocytosis in the invasion process of Streptococcus agalactiae. The authors utilize both novel iPSC-BECs and immortalized hCMEC/D3 cells to demonstrate GBS exposure increases endocytosis and go a step further to identify a potential mechanisms for this uptake, namely the PLC/PKC/Nox2 signaling pathway. The experiments are well designed, controlled, and the results strongly support the conclusions drawn. Minor concerns are outlined below. 

Minor comments:

Introduction:

Line 37 - Recovering “for” should be “from”.

Line 42 - Blood brain barrier shouldn’t be capitalized.

Results:

Line 92 - the term them can be removed to simply say mock infected or infected.

Line 96 - to match the figure, change 1a to 1A. Same for all other panels in this figure and remaining igures.

Nox2 contributes to GBS invasion of BECs, Line 180-181 and 184-186. This paragraph has some confusing portions. The authors continually just refer to BEC’s without designating which data are using iPSC-BECs and which are hCMEC/D3. While it is designated in the titles to the figures in Supplemental, it should be more clearly worded in the results so as to prevent ambiguity. For instance in the results section, Inhibition of PLC decreases GBS invasion, the authors in lines 127-128 state “However, in the immortalized hCMEC/D3 cells, we did not observe…” Similar language should be used throughout the results with switching between data from the different BEC cell types.

Discussion:

Lines 253-256 – These two sentences seem redundant.

Reviewer 2 Report

No obvious flaws or areas of weakness. The conclusion is consistent with the evidence presented.

Reviewer 3 Report

In this manuscript, Espinal et al. report that Group B Streptococcus (GBS), the first etiologic agent for neonatal meningitis, exploits a signalling cascade involving PLC, PKC and Nox2 to invade brain endothelial cells. This study is based on the use of inhibitors to block GBS internalization in IPSC derived brain endothelial cells. Although results from this paper contribute to a better understanding of mechanisms used by GBS to invade cerebral endothelial cells, I have several comments for the authors to consider.

  • Authors claim several times in their manuscript that the PLC, PKC and Nox2 pathway they identify contribute to macropinocytosis. As these 3 molecules can be involved in other endocytic mechanisms than macropinocytosis, authors should reduce emphasizes on the term macropinocytosis and prefer a more generic term such as endocytosis unless they clearly show that GBS internalization is related to macropinocytosis and not Clathrin mediated endocytosis or another mechanism. Macropinocytosis is for instance independent on dynamin-2 and requires small Rho GTPases.

  • It is not clear why authors performed fluid phase uptake of fluorescent dextran in different conditions than the one used to perform GBS internalization assay (5 hours infection in presence of Dextran versus 2 hours infection followed by 2 hrs antibiotic to kill extracellular bacteria for the invasion assay).

  • I would recommend that the authors improve the manuscript by doing a more in-depth analysis of the literature regarding mechanism of GBS internalization in cerebral endothelial cells and others cell types in order to discuss properly their results in regards to what have been published previously. This would greatly improve the discussion as the last part of the discussion on astrocytes pericyte and transporter in the neurovascular unit (lanes 266-272) is not really related to this study and does not bring important insight.

  • The requirement of PLC, PKC and Nox2 in GBS internalization does not mean that they are involved in the same signalling pathway. Yet, the authors claimed lane 183 that they have established a linked pathway between these 3 signalling molecules. This statement should be dampened or clearly experimentally investigated.

Data presented do not always support statement in the text. In addition, authors do not quote any of papers that have previously investigated GBS internalization in cerebral endothelial cells such as:

Maruvada et al. that show involvement of PKCα (as well as Phospholipase A and leukotriens); Zhu et al. that described an internalization pathway involving SphK1/2, S1P, S1P2 receptor, EGFR, PLA2a, CysLT or Deshayes de cambrone et al. showing that GBS internalization involve dynamin-2 and lipid raft but is independent on Pi3Kinase.

They really should discuss it properly as their results suggest that multiple internalization pathways could be involved in cerebral endothelial cells as described for GBS in other cell type (dendritic cells) (Lemire et al.)

  1. Maruvada, R., Zhu, L., Pearce, D., Sapirstein, A. & Kim, K.S. Host cytosolic phospholipase A(2)alpha contributes to group B Streptococcus penetration of the blood-brain barrier. Infect Immun 79, 4088-4093 (2011).
  2. Zhu, N., Zhang, C., Prakash, A., Hou, Z., Liu, W., She, W., Morris, A. & Sik Kim, K. Therapeutic development of group B Streptococcus meningitis by targeting a host cell signaling network involving EGFR. EMBO Mol Med 13, e12651 (2021).
  3. Deshayes de Cambronne, R., Fouet, A., Picart, A., Bourrel, A.S., Anjou, C., Bouvier, G., Candeias, C., Bouaboud, A., Costa, L., Boulay, A.C., Cohen-Salmon, M., Plu, I., Rambaud, C., Faurobert, E., Albiges-Rizo, C., Tazi, A., Poyart, C. & Guignot, J. CC17 group B Streptococcus exploits integrins for neonatal meningitis development. J Clin Invest 131 (2021).
  4. Lemire, P., Houde, M. & Segura, M. Encapsulated group B Streptococcus modulates dendritic cell functions via lipid rafts and clathrin-mediated endocytosis. Cell Microbiol 14, 1707-1719 (2012).

Reviewer 4 Report

The study by Espinal and colleagues addresses whether group B Streptococcus (GBS) uses macropinocytosis as an endocytic pathway to invade BMEC. GBS remains the leading cause of meningitis in newborns. Moreover, the morbidity did not decline over the years, implying that novel approaches are needed to reduce the burden of brain colonization.

Although the authors use a reductionist in vitro approach, they recognize that limitation in the discussion section. The question addressed is well defined, and the presented findings are interesting and novel.

COMMENTS: 

  1. Line 40 “mechanisms behind infection”. Do you mean the mechanism behind brain invasion/infection? Neonatal infection and disease can occur in the absence of meningitis. Please clarify.
  2. Line 41. Although this reviewer recognizes that bacteremia increases the possibility of bacterial interaction with the BBB and thus brain invasion, meningitis can occur without detectable bacteremia. See doi.org/10 .1542/peds.2005-1132. Please rephrase.
  3. Line 77. Please do not report for figures in the introduction section.
  4. ANOVA was used to determine the significance across the different conditions. However, in the figures, it is not clear whether the ** presented is relative only to the comparison with the vehicle group. Please clarify in the legends.
  5. For clarity, the authors should be consistent regarding the reference to the cells used: sometimes, they refer to BECs, iPSC-BECs. Example: line 91 (iPSC-BECs), line 98 (BECs), lines 125-126 (iPSC-BECs), line 130 (BECs) … Please correct throughout the manuscript.
  6. Regarding the experiment presented in Fig. 1. Did the authors check for cell viability? 5h using a MOI 10 with a hypervirulent GBS strain in high. Did they control for extracellular GBS growth?
  7. Commonly, the authors found no differences is the adherence of GBS to BMECs, while the invasion is decreased in the different tested conditions. Do you have any explanation for this observation?
  8. Could this macropinocytosis mechanism of entry into the host cells provide protection from the intracellular degradative lysosomal pathway? Did the authors check for later timepoints when blocking any of the PLC-PKC-Nox2 pathways?
  9. What is mediating the macropinocytosis mechanisms? Could it be the capsule? The authors use a CC17 hypervirulent strain, responsible for most meningitis cases. Would they have the same result with a less virulent strain? For example, using an isogenic non-encapsulated mutant. Please comment on that
  10. The idea that GBS exploits lipid rafts to invade host cells as an infectious strategy has already been reported in the literature. It should be mentioned in the discussion section. Please see org/10.1016/j.ajog.2008.03.051

Methods:

  1. Lines 288-290. Please indicate the brand.
  2. Line 291. For replication purposes, how many cells were seeded? What is the concentration of the coating agents?
  3. Line 307. What antibiotics were used?
  4. Line 310. Treated with dextran how? Dextran concentration? Resuspended in culture medium? At the same time, were they infected with GBS?
  5. Line 317. Verb tense. Were instead of are.

Minor points:

  1. Line 71. I believe there are too many commas.
  2. Sometimes, the figure legends miss the symbol “<” after the P, when indicating the significance. It should be read “P < 0.05” instead of “P 0.05”. Example, figure 1. Please correct throughout the figures' legends.
  3. Lactococcus lactis is a species name and should be in italics (lane 212, 215 and 216)
  4. Figure 3 legend. A and B are written differently from C and D. Please normalize the figure legend when possible.
